# Sodium Intake and Chronic Kidney Disease

**DOI:** 10.3390/ijms21134744

**Published:** 2020-07-03

**Authors:** Silvio Borrelli, Michele Provenzano, Ida Gagliardi, Michael Ashour, Maria Elena Liberti, Luca De Nicola, Giuseppe Conte, Carlo Garofalo, Michele Andreucci

**Affiliations:** 1Nephrology Unit, Advanced Surgical and Medical Sciences Department of University of Campania “Luigi Vanvitelli”, Piazza Miraglia, 80137 Naples, Italy; m.elenaliberti@libero.it (M.E.L.); luca.denicola@unicampania.it (L.D.N.); giuseppe.conte@unicampania.it (G.C.); carlo.garofalo@hotmail.it (C.G.); 2Nephrology Unit, Department of Health Sciences, *“Magna Grecia”* University, 88100 Catanzaro, Italy; michiprov@hotmail.it (M.P.); ida_88@libero.it (I.G.); ashourmichael@yahoo.com (M.A.); andreucci@unicz.it (M.A.)

**Keywords:** salt intake, sodium, hypertension, cardiovascular risk, mortality, prognosis

## Abstract

In Chronic Kidney Disease (CKD) patients, elevated blood pressure (BP) is a frequent finding and is traditionally considered a direct consequence of their sodium sensitivity. Indeed, sodium and fluid retention, causing hypervolemia, leads to the development of hypertension in CKD. On the other hand, in non-dialysis CKD patients, salt restriction reduces BP levels and enhances anti-proteinuric effect of renin–angiotensin–aldosterone system inhibitors in non-dialysis CKD patients. However, studies on the long-term effect of low salt diet (LSD) on cardio-renal prognosis showed controversial findings. The negative results might be the consequence of measurement bias (spot urine and/or single measurement), reverse epidemiology, as well as poor adherence to diet. In end-stage kidney disease (ESKD), dialysis remains the only effective means to remove dietary sodium intake. The mismatch between intake and removal of sodium leads to fluid overload, hypertension and left ventricular hypertrophy, therefore worsening the prognosis of ESKD patients. This imposes the implementation of a LSD in these patients, irrespective of the lack of trials proving the efficacy of this measure in these patients. LSD is, therefore, a rational and basic tool to correct fluid overload and hypertension in all CKD stages. The implementation of LSD should be personalized, similarly to diuretic treatment, keeping into account the volume status and true burden of hypertension evaluated by ambulatory BP monitoring.

## 1. Introduction

Sodium is not only considered an important mineral in maintaining the balance of body fluid, but it has also played an important role in the history of the world for its economical, religious, and symbolic importance. The concept that salt is a beneficial substance was so ingrained that although the earlier studies on the relationship between low sodium intake and reduction of blood pressure (BP) date back to 1948 [1], it was only after nearly forty years that the international community has recognized the role of salt intake in the pathophysiology of hypertension [2]. According to the World Health Organization, the restriction of sodium intake to less than 2.3 g/day of sodium corresponding to 5.8 g of salt (or 100 mmol) is one of the most cost-effective measures to improve public health [3]. Cumulating evidence highlights that higher sodium consumption contributes to higher BP [4], thus increasing the risk of cardiovascular disease (CVD) [5,6]. However, recent studies have raised some concerns about the real benefit of a low salt diet in the healthy general population [7,8,9,10]. In particular, in a large cohort study in over 100,000 patients from 18 countries the role of higher salt consumption was associated with increased BP levels [7], and poor CV outcomes [8]. At the same time, it emerged that sodium intake of <3 g forecasted a higher CV risk, drawing a U-shaped mortality curve [8]. Although these findings have also been obtained by other investigators [8,9,10], these studies are methodologically flawed by reverse causality (e.g., patients eat less salt because they are sicker and/or more malnourished), collinearity (e.g., a low sodium intake may be associated with a low protein-energy intake) and biased methods used to assess individual salt intake (e.g., single spot urine sample) [11].

In Chronic Kidney Disease (CKD) patients, high BP is a frequent finding, which is traditionally considered as a direct consequence of sodium sensitivity. Hence, a low salt diet (LSD) is widely considered a cornerstone in the treatment of hypertension in CKD.

In this review, we address the importance of the kidney in sodium regulation, the relationship of sodium intake with hypertension from earlier CKD stages to end-stage kidney disease (ESKD), and the available evidence on the benefit of salt restriction in non-dialysis CKD and in the ESKD population.

## 2. Adherence to Low-Salt Diet: Definition and Assessment in CKD Patients

The terms sodium and salt (generally sodium chloride) are used interchangeably, generating confusion about sodium intake. Table 1 illustrates formulas to convert sodium in salt (sodium chloride) and *vice versa*, according to the units of measurement.

The first concern in the evaluation of adherence to LSD is the method used for evaluation of sodium intake. Accordingly, the measurement of sodium excretion by 24 h urine sample collection (UNaV) is considered as the gold standard. However, UNaV may be cumbersome for the patient, and, therefore, estimation from spot urine samples using the Nerbass, Kawasaki, Tanaka, and INTERSALT formulas have been proposed to evaluate sodium intake. The rationale arises from the assumption that spot urine excretion would be proportionate to UNaV corrected for creatinine excretion. However, a cross-sectional study in CKD patients showed that these formulas might provide an inaccurate estimate of sodium intake, irrespective of severity of CKD and use of diuretics [12].

Another issue is the evaluation of salt intake by a single measurement, which is not generally considered sufficient to evaluate an individual’s usual salt intake because of the wide day-to-day variability in salt consumption and urinary excretion [13].

Among CKD cohorts, the Chronic Renal Insufficiency Cohort (CRIC) study reported that only about one out of four patients had a sodium intake <100 mmol/24 h, evaluated by three measurements [14]. As reported in Table 2, these findings are consistent with the prevalence of LSD reported in secondary analyses of trials [15,16,17,18,19], and also when CKD patients were regularly followed in nephrology clinics (<25% had a salt intake below 6 g/day) [20].

Novel methods for self-monitoring of salt intake have been developed to improve the adherence to the LSD; these are based on urine chloride strips, given that urinary chloride excretion is very tightly correlated with urinary sodium excretion. The potential benefit of self-monitoring is the ability to immediately achieve an adequate estimate of sodium intake immediately (75.5% sensitivity and 82.6% specificity to correctly classify patients with UNaV >100 mmol/24 h) in order to make proper dietary adjustments aimed at achieving recommended intake [23]. However, any benefit on the achievement of the BP goal with use of chloride strips has still to be proved with use of chloride strips. To answer this question, one randomized clinical trial, the SALUTE-CKD (SALt lowering by Urine sodium self-measurement Trial in Chronic Kidney DiseasE) has advanced to the final stage of development and results are expected in the next few months.

Furthermore, a recent trial performed in 99 patients has evaluated the efficacy of a web-based self-management program for dietary sodium restriction compared with routine care. After 3 months in intervention group a significant reduction of sodium intake (−40 mmol/day) and systolic BP (−8 mmHg) was registered in the intervention group, whereas no significant difference was found in control group. Surprisingly, in the following maintenance phase, no difference in sodium intake was detected between the two groups, due to the inadvertent adoption of the intervention by the control group. Notably, the largest effect was reported in the first 3 months, when participants actively used the web-based self-management program [24].

In ESKD patients, sodium intake can be estimated with a dietary questionnaire, though several factors, such as high dialysate sodium concentration and sodium plasma concentration, can affect thirsty and water intake in these patients, irrespective of their sodium intake [25].

Finally, in ESKD patients, residual kidney function must be carefully evaluated: in this subgroup of patients, dialysis is started with an incremental approach, corresponding to a low dose of dialysis (peritoneal or hemodialysis) integrated into the conservative management [26,27]. In these patients, the assessment of sodium intake by UNaV may be misleading, because of the aliquot of sodium intake removed by dialysis.

## 3. Hypertension and Salt in CKD

Hypertension and CKD are common chronic noncommunicable diseases strictly inter-related with each other; indeed, elevated BP is not only a frequent complication of CKD [28], but it can also act as the cause of CKD [29]. A recent meta-analysis showed that hypertensive patients have a 75% greater risk than normotensive individuals of development of *de novo* CKD (GFR <60 mL/min/1.73 m^2^), estimating a 10% increase of CKD onset for each increase of 10 mmHg of either BP component. Notably, even pre-hypertension (Systolic BP of 120–139 mm Hg and/or Diastolic BP of 80–89 mm Hg) was associated with a 25% higher risk of developing low GFR [29].

Furthermore, the prognostic role of lowering BP assumes greater importance in CKD patients if we bear in mind at least three basic points: (1) higher prevalence of hypertension in CKD than in the general population, which increases progressively from 65% to 95% as GFR falls from 85 to 15 mL/min/1.73 m^2^ [29]; (2) hypertension is the main known risk factor for CKD progression and for CV mortality [30]; (3) Hypertension is often resistant to the treatment in CKD patients, resulting in worsening CV prognosis [31,32].

Salt and water retention play a key role for development of hypertension in CKD. In fact, according to the classical model, under normal conditions, high salt intake temporarily increases plasma sodium level, which is soon buffered by movement of water from the intracellular to the extracellular compartment. Thus, increased plasma sodium concentration also stimulates the thirst center, leading to an increase in water intake and secretion of antidiuretic hormone, which restores plasma sodium concentration to a normal level while increasing and maintaining extracellular fluid volume. On the other hand, high salt intake suppresses the renin-angiotensin-aldosterone system (RAAS), which consequently reduces sodium tubular reabsorption, thus contributing to re-establishing sodium and water homeostasis [33].

In CKD patients, external sodium balance is preserved by expansion of the extracellular volume (ECV), which however causes the persistence of high BP levels. Therefore, hypertension in CKD is an early manifestation of ECV expansion and, at the same time, a maladaptive mechanism aimed at limiting ECV expansion that corresponds to approximately 5% to 10% of body weight, generally without peripheral edema, when cardiac and hepatic function is normal and the transcapillary Starling forces are not disrupted [34]. In spite of ECV expansion, RAAS is inappropriately activated in CKD, leading to vasoconstriction and sodium retention, which contribute significantly to the raising of BP levels [35].

As reported in a classic experiment [36], the BP response to sodium load is amplified in CKD patients. In particular, increasing sodium intake was increased from 20 to 120 mmol/day in patients with advanced renal failure, this caused a significant acute increase of BP (+12.2 ± 1.4 mmHg). On the other hand, the same increase in sodium intake in healthy people was not associated with any BP change and, even greater elevation of sodium intake up to 1120 mmol/day, did not produce any effect on BP values. This experiment is the proof of concept of the sodium sensitivity of BP in CKD. Notably, sodium sensitivity may be already detectable in the earlier CKD stages, as reported in a study comparing patients with glomerular disease vs healthy controls, which showed a significant BP reduction in response to lowering salt intake, whereas BP did not change in controls [37].

Moreover, experimental studies showed that high salt intake induces intrarenal production of Angiotensin-II [38], stimulates the synthesis of pro-inflammatory cytokines [39] and increases oxidative stress [40], as well as triggering sympathetic activity [41], whose activation is already increased in CKD, as a result of increased arterial stiffness and/or endothelial dysfunction [42].

## 4. Alternative Mechanism of Sodium Toxicity

Recent experimental findings suggest that skin could work as a reservoir of sodium, escaping from renal control [43]. In particular, high salt intake might cause sodium accumulation in the skin, which is detected by cells of the Monocytes Phagocytes System (MPS) located in the skin interstitium, which act as osmoreceptors by expression of the tonicity enhancer-binding protein (Ton-EBP). This transcription factor leads to Vascular Endothelial Growth Factor (VEGF) production that increases sodium clearance by the lymphatic network [44,45]. Moreover, high sodium levels in the CKD condition would promote the expression of pro-inflammatory factors, such as Interleukin-6, VEGF, and Monocyte Chemoattractant Protein-1 (MCP-1), via Ton-EBP pathway, leading to local inflammation and vascular proliferation in peritoneal, heart, and vascular tissue [46].

Figure 1 summarizes the potential mechanisms underlying the increase of BP levels and dependent CV risk associated with high salt intake in CKD.

The concept that sodium balance is regulated by additional extra-renal mechanisms was first reported by Herr et al., whose study showed that high salt intake increased total sodium content, whereas total body water and body weight did not change [47]. More recently, a space flight simulation study has reported that in healthy subjects under controlled sodium intake, UNaV changes periodically (every 6 days), independently from BP levels and total body water [48].

The recent availability of ^23^Na Magnetic Resonance Imaging (MRI) in humans has allowed detection and quantification of sodium storage in the skin [49]. In particular, a higher tissue sodium content was detected in patients affected by hyperaldosteronism. Interestingly, surgical and/or medical correction of hyperaldosteronism was associated with a significant reduction in tissue sodium content; whereas body weight did not change [50]. Recently, in a cross-sectional analysis of 99 CKD patients, skin sodium content was strongly associated with left ventricular mass independently from BP levels and volume status [51]. Finally, sodium stored into the skin is modifiable in CKD patients, as reported by a recent study showing a significant reduction of skin sodium content, after a single hemodialysis session, though the mechanism by which sodium is removed from skin remains still unclear [52].

## 5. Clinical Effects of Low Salt Diet in Non-Dialysis CKD

We have recently completed a metanalysis comparing low versus high salt diet in 738 CKD patients [53]. Analysis included nine trials [54,55,56,57,58,59,60,61,62]. This meta-analysis showed that a moderate salt restriction of 4.4 g/day (from 179 mEq/day to 104 mEq/day) was associated with a significant lowering of 4.9 mmHg [95% C.I.: 6.8/3.1 mmHg; *p* < 0.001] in systolic BP and of 2.3 mmHg [95% C.I.: 6.8/3.1 mmHg; *p* < 0.001] in diastolic BP measured by traditional method [53]. A similar effect was found in the five out of eleven studies [57,60,61,63,64] evaluating the effect of LSD on Ambulatory BP (ABP). In particular, we found that salt restriction reduces systolic and diastolic ABP of 5.9 mmHg (95% C.I.: 2.3/9.5 mmHg; *p* < 0.001) and 3.0 mmHg (95% C.I.: 1.7/4.7 mmHg; *p* < 0.001), respectively [53].

As regards ABP studies, it is worth mentioning that in CKD cohorts, sodium sensitivity has been associated with a higher prevalence of altered circadian rhythm and nocturnal hypertension [65,66], which are predictors of poor cardio-renal prognosis [67].

Moreover, in seven out of eleven studies [54,55,56,57,58,59] reporting the effect of salt restriction on proteinuria, pooled analysis showed a significant improvement of 0.4 g/day (95% C.I.: 0.2–0.6 g/day) associated with lower salt intake [53]. These findings are in agreement with a previous meta-analysis reporting that in patients following a lower salt diet, there was an augmented antiproteinuric effect of RAAS blockers [68]. The synergic effect of LSD and RAAS inhibition may be correlated to the finding that high salt intake enhances angiotensin-converting enzyme (ACE) activity in renal tissues, in spite of decreased plasma renin and angiotensinogen concentrations, which could reduce the effect of RAAS blockers in tissues [38].

Although these effects of LSD on BP and proteinuria suggest an improvement of prognosis in CKD patients, few studies [14,15,16,17,18,19] have evaluated the long-term effect of salt restriction on the cardio-renal outcomes (Table 2).

In the CRIC study, a large observational study carried out in 3757 CKD patients followed for almost seven years, the group of patients with a UnaV of >195 mmol/day was associated with a higher risk of CKD progression [14]. Among participants of this study, 804 composite CV events (575 heart failure, 305 myocardial infarctions, and 148 strokes) occurred during a median of 6.8 years of follow-up, drawing a linear relation between higher sodium intake and higher CV risk [22]. Similarly, a post-hoc analysis of the Reduction of Endpoints in NIDDM with the Angiotensin II Antagonist Losartan (RENAAL) and Irbesartan Diabetic Nephropathy Trial (IDNT) trials in a subgroup of 1177 patients with available 24 h urinary sodium measurements, showed that the beneficial effects of RAAS blockers on renal and cardiovascular outcomes were greater in patients with lower sodium intake [18]. Furthermore, in Autosomal Dominant Polycystic Kidney Disease (ADPKD) patients, fast progressors irrespective of intensive CKD management [69,70], a recent post-hoc analysis of the HALT-PKD trial has shown that a moderate salt restriction reduces CKD progression [21].

On the other hand, other studies have not confirmed these results, finding no association between low salt intake and improvement of the renal prognosis, in CKD patients [15,16,17]. In particular, secondary analysis of the first and second Ramipril Efficacy in Nephropathy (REIN) trials showed that low salt intake was associated with a lower risk of ESKD, but this association disappeared after adjustment for basal proteinuria [15]. In the longitudinal follow up of the Modification of Diet in Renal Disease (MDRD) Study, no association of single baseline 24 h urinary sodium excretion with kidney failure and a composite outcome of kidney failure or all-cause mortality was found [16]. Similarly, post-hoc analysis of the ongoing telmisartan alone and in combination with ramipril global endpoint trial (ONTARGET) and telmisartan randomized assessment study in ACE intolerant subjects with cardiovascular disease (TRANSCEND) studies trials showed no association between UNaV (though estimated by morning spot urine) and renal endpoints (30% decline of eGFR or ESKD) in patients with or without CKD at baseline [17]. Surprisingly, in diabetic non-CKD patients, UNaV was inversely associated with a cumulative incidence of ESKD, and in fact, patients with the lowest sodium excretion had the highest cumulative incidence of ESKD [19].

Of note, the negative studies are post hoc analyses of clinical trials designed to test the efficacy of RAAS inhibitors rather than of low-sodium intervention, confounding thus a possible association [15,16,17,18]. Furthermore, in some of these studies, UNaV was measured by a single 24 h urine [19] or spot urine sample [17]. On the other hand, we cannot exclude that other factors might play a role: a renal hemodynamic response to an acute reduction of sodium intake was impaired by aging, especially when atherosclerotic damage coexists [71]. This may expose patients to acute kidney injury and hypotension [72]. Furthermore, patients with CKD have a higher prevalence of white coat effect [73,74], exposing CKD patients to “inappropriate” antihypertensive treatment, which may potentially cause renal hypoperfusion [75]. Therefore, particular attention must be paid in the management of CKD patients, personalizing salt intake on the basis of “true” hypertensive status measured by ABPM and volemic status, and monitoring the adherence and anti-hypertensive effect LSD over time.

## 6. Sodium Intake in End-Stage Kidney Disease

In ESKD patients, similarly to early CKD stages (Figure 1), the deleterious effects of high salt intake are mainly related to the fluid overload, resulting in high BP levels, left ventricular hypertrophy, and increased CV mortality [76,77,78,79,80]. Therefore, sodium restriction is a major therapeutic goal in these patients. Indeed, it has been estimated that, in ESKD patients with no residual diuresis, a salt intake of <6 g should cause patients to gain no more than 0.8 kg/day in interdialytic weight.

A recent metanalysis of four trials (3 in HD/1 in PD) showed that ESKD patients with lower salt intake (N = 67) had a significant improvement of both systolic [−8.4 (−12.0; −4,8) mmHg] and diastolic BP [−4.4 (−6.6; −4.2) mmHg] levels compared with the higher salt intake group (N = 64) [81]. Moreover, a post-hoc analysis of the HEMO study revealed that low sodium intake, evaluated by a 24 h food questionnaire, allowed to decrease the need for ultrafiltration, even if it was not associated with pre-dialysis systolic BP levels [82].

Similarly, hypervolemia is prevalent in Peritoneal Dialysis (PD) patients, because of the common mismatch between intake and removal of sodium and fluid [83,84,85]. In a recent study performed in a cohort of 1054 incident PD patients, overhydration was evident in over 50% of patients starting PD [83]. This finding is relevant because persistence of volume overload heralds a 60% higher mortality risk [84]. Interestingly, recent experimental findings have reported that high sodium intake is related to direct toxicity on the peritoneal membrane, leading to chronic inflammation, fibrosis, and hypervascularization, increasing, in turn, peritoneal permeability [86].

Surprisingly, few studies have addressed the relationship between sodium intake and mortality in ESKD patients (Table 3). In hemodialysis, secondary analysis of HEMO study showed that higher dietary sodium correlated with mortality rate independently from patients’ nutritional status [82]. A retrospective study on 305 Chinese PD patients has reported that sodium intake, assessed by a 3-day diet questionnaire, was inversely associated with all-cause and cardiovascular mortality. It is noteworthy that patients with lower sodium intake also had lower serum albumin levels and reduced lean body mass, as well as lower energy and protein intake, when compared with patients with higher sodium intake, suggesting that patients with a lower dietary sodium intake were more malnourished and with reduced appetite. Moreover, the group of patients with highest salt intake had mean sodium intake of 2.5 g/day, which is lower than the mean intake reported in the general population, suggesting a possible measurement bias. This study, moreover, was flawed by methodological issues (small sample size, monocentric, few events, and overfitted cox models), further reducing the generalizability of the results [87].

Therefore, salt restriction remains the basic approach to achieve volume control in PD patients, keeping in mind that sodium removal is lower in PD patients treated with cycler (Automated PD, APD), because of greater sodium sieving as compared with Continuous Ambulatory PD (CAPD). In these patients, high salt intake may not be counterbalanced by sodium removal, consequently leading to hypervolemia and hypertension [88].

## 7. Conclusions

The negative effects of sodium on BP values are amplified in CKD patients, as a result of fluid overload and of direct toxicity on the heart, the vascular system, and kidney. In non-dialysis CKD patients, LSD is beneficial for hypertension control, irrespective of BP levels, to lower proteinuria by enhancing the antiproteinuric effect of RAAS inhibition. Whether these effects can improve cardio-renal prognosis still remains unclear. Nonetheless, salt restriction assumes a greater importance in ESKD because of the common mismatch between intake and removal of sodium, which leads to hypertension, LVH, and higher CV risk. Therefore, reducing salt intake is crucial for hypertensive CKD patients from earlier stages to ESKD. However, it remains insufficiently and/or inadequately applied. More studies are therefore needed to improve adherence to LSD in the long term.

## Figures and Tables

**Figure 1 ijms-21-04744-f001:**
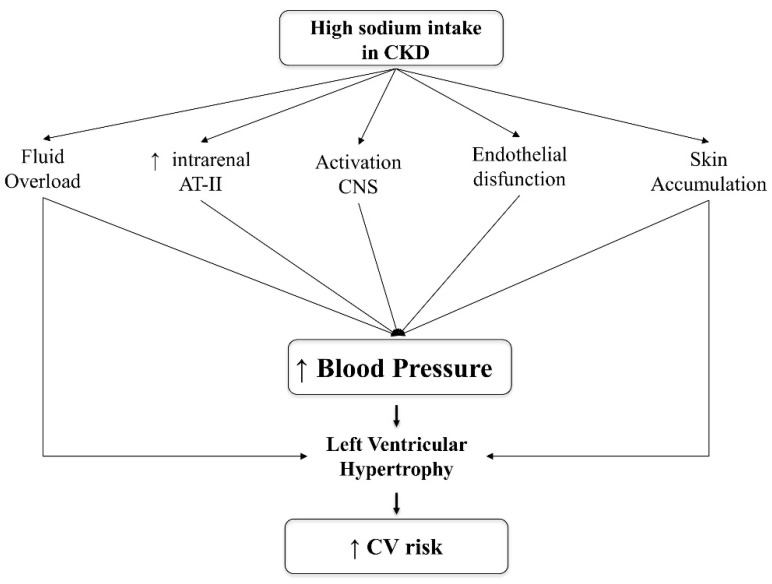
Potential pathogenic mechanisms of hypertension in CKD due to high salt intake. Abbreviations: CKD: Chronic Kidney Disease; AT-II: Angiotensin-II; CNS; Central Nervous System; CV: cardiovascular.

**Table 1 ijms-21-04744-t001:** Formulas to convert sodium in salt (sodium chloride) and vice versa, according to units of measurement.

grams of sodium = mmol of sodium × 0.023
grams of salt = mmol of sodium × 0.058 (or mmol of sodium/17)
grams of sodium = grams of salt × 0.394
grams of salt = grams of sodium × 2.542

**Table 2 ijms-21-04744-t002:** Studies evaluating the effect of urinary sodium excretion (UNaV) on end-stage kidney disease (ESKD) and cardiovascular (CV) outcomes in patients with or without Chronic Kidney Disease (CKD).

Author [ref], Year	Type	Sample Size	Mean eGFR	Method to Assess UNaV	Prevalence LSD	ESKD	CV Outcomes
Torres, V.E. [21], 2017	Post-hoc analysis HALT-PKD trial (Study B)	486	48.6	Multiple 24 h urine sample	n.a.	Averaged UNaV is associated with increased risk for the combined endpoint of death, ESRD or 50% eGFR decline [HR 1.08 (1.01–1.06) per 18 mmol/day]	n.a.
He, J. [14], 2016	Longitudinal prospective CRIC study	3757	43.4	Three 24 h urine sample	27.7% (<116 mmol/day)	Highest quartile of UNaV (>194.6 mmol/day) is associated with higher risk of ESKD [HR: 1.46 (1.09–1.70)]	n.a.
Mills, K.T. [22], 2016	Longitudinal prospective CRIC study	3757	43.4	sodium/creatinine ratio from multiple 24 h urine samples	25.0% (<2894 mg/g)	n.a.	Highest quartile of calibrated UNaV (>4548 mg/g) is associated with higher risk of composite CV endpoints [HR: 1.36 (1.09–1.70)]
Fan, L. [16],2014	Post-hoc analysis MDRD trial	840	32.5	Three or four 24-h urine sample	25.0% (<93 mmol/day)	No association was found between mean baseline UnaV and ESKD [HR 0.99 (95% CI 0.91–1.08)]	n.a.
Smyth, A. [17], 2014	Post-hoc analysis ONTARGET TRANSCEND trials	28,879	68.4	Single fasting urine sample	2.7% (<87 mmol/day)	There was no association between estimated UNaV and risk of renal outcomes (ESKD or 30% eGFR decline)	n.a.
Vegter, S. [15], 2012	Post-hoc analysis of REIN1 and -2 trials	500	43.2	sodium/creatinine ratio from multiple 24 h urine samples	22.2% (<100 mmol/day)	Unadjusted analysis showed association between UNaV and renal outcome, which disappears after adjustment for proteinuria	n.a.
Lambers Heerspink, H.J. [18], 2012	Post-hoc analysis of RENAAL and IDNT trials	1177	44.0	sodium/creatinine ratio from multiple 24 h urine samples	33.3% (<121 mmol/day)	In the group of patients treated with ARBs, thelowest UNaV tertiles of were associated with improved renal outcome [HR: 0.57 (0.39–0.84)]	In the group of patients treated with ARBs, the lowest UNaV tertiles were associated with improved CV outcome [HR: 0.54 (0.34–0.86)]
Thomas, M.C. [19], 2011	Longitudinal prospective Study	2807	n.a.	a single 24 h urine collection	25% (<102 mmol/day)	In diabetic patients, UnaV was inversely associated with the cumulative incidence of ESRD.	n.a.

Abbreviations: n.a.: not applicable. ARBs: Angiotensin Receptor Blockers.

**Table 3 ijms-21-04744-t003:** Studies evaluating the effect of urinary sodium excretion (UNaV) on mortality in ESKD patients.

Author [ref], Year	Type	Sample Size	Dietary Sodium Evaluation	Outcomes
*Hemodialysis*				
McCausland [82], 2012	Post-hoc analysis of HEMO trial	1170	48 h food diary	Higher dietary salt intake is associated with greater mortality.
*Peritoneal Dialysis*				
Dong [87], 2010	Retrospective analysis	305	3-day dietary records	Higher dietary salt intake is associated with lower mortality (aHR:0.45 (0.23–0.90)

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
