# Peer review of "Sodium Intake and Chronic Kidney Disease"

_ijms, 2020, doi:10.3390/ijms21134744_

Round 1
Reviewer 1 Report
This is a straightforward review about a very important topic.
I think only a minor revision is necessary:
1) the first reference should be the fundamental paper by Walther Kempner on the Bull N Y Acad Sci, 1949.
2) The table 1 has some rounding mistakes: 2nd row "x 0.058" instead of "x 0.059" (the actual value is 0.05842)
3rd row "x 0.394" instead of "x 0.393" (the actual value is 0.3937)
4th row "x 2.54" instead of "x 2.55" (the actual value is 2.5421)
3) row 61 at page 6: the word "increased" should be deleted
4) Throughout the reference section the "doi" codes should be added as per the author's guide.
Author Response
We thank Reviewer 1 for her/his comments, that allowed to improve our manuscript.
We have accepted all the changes suggested by him/her.
Following point-by-point response:
1) the first reference should be the fundamental paper by Walther Kempner on the Bull N Y Acad Sci, 1949.
We have added the reference, accordingly.
2) Table 1 has some rounding mistakes: 2nd-row "x 0.058" instead of "x 0.059" (the actual value is 0.05842)
3rd row "x 0.394" instead of "x 0.393" (the actual value is 0.3937)
4th row "x 2.54" instead of "x 2.55" (the actual value is 2.5421)
We have corrected Table 1, as suggested by Reviewer 1.
3) row 61 on page 6: the word "increased" should be deleted
We have deleted it.
4) Throughout the reference section the "doi" codes should be added as per the author's guide.
We have added the "doi" per each reference.
Reviewer 2 Report
Timely and appropriate subject for discussion on the perils of excess sodium intake in CKD /ESRD and associated CV morbidity and mortality.
However, the paper would benefit from an extensive editing and correction of grammar and syntax.
Author Response
We thank Reviewer2 for his/her comment, which allowed us to improve our manuscript.
As required, we have performed extensive English editing.
Round 2
Reviewer 2 Report
There are a few minor editing required:
- Introduction Line 63. "In this review, we address the importance of the kidney in the sodium regulation,---“ . ‘the’ should be removed
- Discussion Line 20: “actively used actively the web based self-management program”. The second actively should be removed
Line 24: “Finally, in ESKD patients with residual diuresis must be ----“. Authors need to clarify. Diuresis is increased production of urine, hence cannot be a residual. ‘Residual kidney function” may be more appropriate
Line 36: Diastolic BP of was associated with a 25% risk higher of developing low GFR. This should be “higher risk”
Line 41-42; ------ hypertension is often resistant to the treatment in CKD patients, furtherly worsening their CV prognosis. Please rephrase the sentence
Line 44; “According to the classical model, in fact under normal conditions------“. Please remove ‘in fact’
Line 57: “ limiting ECV expansion that it corresponds to” Pls delete ‘it’
Line 69: ” ------ in glomerular patients with and without CKD vs health” . Please rephrase sentence as all patients have glomeruli. Change health to ‘healthy’
Line 79: ‘----------- cells of the Monocytes Phagocytes System (MPS) cells located ----“ Please delete the second ‘cells’
Line 86: “The figure 1 summarizes the -------“ . Pls change to ‘ Figure 1’
Line 94: “The concept------ mechanism was firstly”. Pls change to firstly to ‘first’
Line 103: ‘Interestingly, when surgical and /or medical -----“. Please delete ‘when’
Line 126: “ Moreover, ------ salt restriction diet on”. Pls delete ‘diet’
Line 153 : ” On the other hand, ------- these results by disclosing no significant”. Confusing statement. Please rephrase sentence
Line 190-192: “ A post hoc analysis ----------- allowed the decrease of UF requirement, while being not associated with pre-dialysis systolic BP levels” . Please rephrase sentence
Line 224-5: “ The negative effects of ------------ patients, as result of fluid overload-------“. Should be ‘as a result of ’
Author Response
Response to reviewers:
Introduction Line 63. "In this review, we address the importance of the kidney in the sodium regulation,---“ . ‘the’ should be removed
R. We deleted the term “the” from this sentence.
Discussion Line 20: “actively used actively the web based self-management program”. The second actively should be removed
R. Yes correct. The second “actively” has been deleted.
Line 24: “Finally, in ESKD patients with residual diuresis must be ----“. Authors need to clarify. Diuresis is increased production of urine, hence cannot be a residual. ‘Residual kidney function” may be more appropriate
R. We have changed this sentence accordingly by replacing residual diuresis with Residual kidney function.
Line 36: Diastolic BP of was associated with a 25% risk higher of developing low GFR. This should be “higher risk”
R. Yes, thank you. We have anticipated “higher” before “risk”
Line 41-42; ------ hypertension is often resistant to the treatment in CKD patients, furtherly worsening their CV prognosis. Please rephrase the sentence
R. We have changed the sentence by stating: “Hypertension is often resistant to the treatment in CKD patients, resulting in worsening CV prognosis”
Line 44; “According to the classical model, in fact under normal conditions------“. Please remove ‘in fact’
R. We have deleted “in fact” from the mentioned phrase
Line 57: “ limiting ECV expansion that it corresponds to” Pls delete ‘it’
R. We have deleted “it” from the sentence, accordingly.
Line 69: ” ------ in glomerular patients with and without CKD vs health” . Please rephrase sentence as all patients have glomeruli. Change health to ‘healthy’
R. We revised this sentence by stating “..study comparing patients with glomerular disease vs healthy controls”
Line 79: ‘----------- cells of the Monocytes Phagocytes System (MPS) cells located ----“ Please delete the second ‘cells’
R. We have deleted the second “cells” from the sentence.
Line 86: “The figure 1 summarizes the -------“ . Pls change to ‘ Figure 1’
R. We changed “The figure 1..” to “Figure 1”
Line 94: “The concept------ mechanism was firstly”. Pls change to firstly to ‘first’
R. We changed “firstly” with “first”.
Line 103: ‘Interestingly, when surgical and /or medical -----“. Please delete ‘when’
R. Correct! We deleted “when” from the sentence.
Line 126: “ Moreover, ------ salt restriction diet on”. Pls delete ‘diet’
R. We deleted “diet” from that phrase.
Line 153 : ” On the other hand, ------- these results by disclosing no significant”. Confusing statement. Please rephrase sentence
R. We revised the sentence: “On the other hand, other studies have not confirmed these results, finding no association between low salt intake and an improvement of the renal prognosis, in CKD patients”
Line 190-192: “ A post hoc analysis ----------- allowed the decrease of UF requirement, while being not associated with pre-dialysis systolic BP levels” . Please rephrase sentence
R. We agree with the reviewer that this sentence was not completely clear. Moreover, the acronym UF (ultrafiltration) was not previously specified in the manuscript. Hence, we have now changed the sentence in: “Moreover, a post-hoc analysis of the HEMO study revealed that low sodium intake, evaluated by a 24h food questionnaire, allowed to decrease the need for ultrafiltration, even if it was not associated with pre-dialysis systolic BP levels”
Line 224-5: “ The negative effects of ------------ patients, as result of fluid overload-------“. Should be ‘as a result of ’
R. We have changed the sentence following this suggestion.